# Clients satisfaction at primary healthcare facilities and its association with implementation of client service charter in Tanzania

Erick S. Kinyenje[1]*, Talhiya A. Yahya[1], Mbwana M. Degeh[1], Chrisogone C. German[1], Joseph C. Hokororo[1], Mohamed A. Mohamed[2,3], Omary A. Nassoro[1], Radenta P. Bahegwa[1], Yohanes S. Msigwa[1], Ruth R. Ngowi[1], Laura E. Marandu[1], Syabo M. Mwaisengela[4], Eliudi S. Eliakimu[1]

**1** Health Quality Assurance Unit, Ministry of Health, Community Development, Gender, Elderly and Children, Dodoma, Tanzania, **2** Tanzania Field Epidemiology and Laboratory Training Programme (TFELTP), Dar es Salaam, Tanzania, **3** East Central and Southern Africa Health Community, Arusha, United Republic of Tanzania, **4** Regional Administrative Secretary's Office—Regional Health Management Team, Mtwara, Tanzania

* kinyenje2003@yahoo.com

**Data Availability Statement:** Permission to access the data was granted by Ministry of Health Tanzania under the Health Quality Assurance Unit

## Abstract

### Background

Client service charter (CSC) provides information about what people can expect in a facility's services; what is expected of clients and service providers. Tanzania implemented Star Rating Assessment (SRA) of primary health care (PHC) facilities in 2015/16 and 2017/18 using SRA tools with 12 service areas. This paper assesses the status of service area 7, namely client focus that checked if client was satisfied with services provided and implementation of CSC through three indicators–if: CSC was displayed; CSC was monitored; client feedback mechanism and complaints handling was in place.

### Methods

We extracted and performed a cross-sectional secondary data analysis of data related to clients' focus that are found in national SRA database of 2017/2018 using STATA version 15. Client satisfaction was regarded as dependent variable while facility characteristics plus three indicators of CSC as independent variables. Multivariate logistic regression with p-value of 5% and 95% confidence interval (CI) were applied.

### Results

A total of 4,523 facilities met our inclusion criteria; 3,987 (88.2%) were dispensaries, 408 (9.0%) health centres and 128 (2.8%) hospitals. CSC was displayed in 69.1% facilities, monitored in 32.4% facilities, and 32.5% of the facilities had mechanisms for clients' feedback and handling complaints. The overall prevalence of clients' satisfaction was 72.8%. Clients' satisfaction was strongly associated with all implementation indicators of CSC. Clients

(third party). The author did not have the right to share the data publicly. In order to gain access to data kindly contact: ps@afya.go.tz, operated by the office of the chief executive officer of the Ministry.

**Funding:** The author(s) received no specific funding for this work.

**Competing interests:** We have read the journal's policy and the authors of this manuscript have the following competing interests: During the time of baseline and reassessment as well as during write up of the study – JH, EE and TY were with the Health Quality Assurance Divison (now called Health Quality Assurance Unit) and were responsible for the implementation of SRA and QIPs folllow-up. However, this does not alter our adherence to PLOS ONE policies on sharing data and materials.

from urban-based facilities had 21% increased satisfaction compared rural-based facilities (AOR 1.21; 95%CI: 1.00–1.46); and clients from hospitals had 39% increased satisfaction compared to dispensaries (AOR 1.39; 95%CI: 1.10–1.77).

## Conclusion

The implementation of CSC is low among Tanzanian PHC facilities. Clients are more satisfied if received healthcare services from facilities that display the charter, monitor its implementation, have mechanisms to obtain clients feedback and handle complaints. Clients' satisfaction at PHC could be improved through adoption and implementation of CSC.

## Introduction

The Astana Declaration emphasizes empowering individuals and communities through their participation in the development and implementation of policies and plans that impact health; protection and promotion of solidarity, ethics and human rights [1]. Unlike the Alma-Ata declaration held 40 years earlier (1978); the Astana declaration addresses the wider scope of primary healthcare, the scope that goes beyond building primary healthcare systems by considering current health challenges, embracing universal health coverage initiatives and political involvement [2].

Clients attending primary health care (PHC) facilities have been experiencing inadequate quality services, especially in the aspects of interpersonal quality in terms of communication. For example, the analysis of the data in demographic and health surveys and the service provision assessment surveys which aimed at comparing the quality of care in public hospitals and health centres, has found that "*hospital users were more likely to report experiencing problems with the amount of explanation received from their provider and with their ability to discuss concerns [3]*. Also, more clients (81.3%) in health centres, reported being very satisfied with the services received, compared with 74.7% in hospitals [3].

This paper discusses service area number seven (7) of the star rating assessment (SRA) tools, namely client focus. Area 7 is one of 12 areas of the SRA tool; others are Legality (Licensing and Certification), Health Facility Management, Use of Facility Data for Planning and Service Improvement, Staff Performance Assessment, Organization of Services, Handling Emergencies and Referral, Social Accountability, Facility Infrastructure, Infection Prevention and Control (IPC), Clinical Services, and Clinical Support Services [4, 5]. The performance from the 12 areas from each PHC facility are aggregated and computed to form a Star that could be assigned to that facility; as described in detail by Talhiya Y and Mohammed M [4].

The client charter is the document that provides information about what people can expect from their treatment in these services. It also outlines what is expected of clients and the service providers [6]. Client charter also helps to nudge health workers to provide services in a way that upholds ethics and observes patient rights in their interaction with patients [7, 8]. In situations where client's rights are not well disseminated and known by citizens, patients attending primary health care facilities may not be able to complain or voice a concern [9]. In Uganda, a study involving a network of faith-based facilities found that out of all client satisfaction dimensions assessed, the dimension of rights scored low due to "*dissatisfaction of clients in the way health providers engaged with them in informing them of their entitlements and in the decision-making process*" [10]. The Tanzanian client charter includes a list of services the facility provides like paediatrics, obstetrics, gynaecology, types of insurances allowed, types of tests/

investigation and time of services provision, e.g., daytime or throughout the day (i.e., 24/7). From a star rating perspective, the client charter aimed to look at the following: if client services charter was displayed; client services charter was monitored; client feedback mechanism and complaints handling was in place; and checking if the client was satisfied with services provided by the health facility, as shown in **Table 1**.

Implementation of the client service charter can increase accountability in health facilities by stimulating actions from both supply–and demand sides [11]. Implementation of patient's rights as stipulated in client service charter is also aligned with the efforts to improve the quality of health services delivery based on the fact that the listed rights match with the dimensions of quality [12–15]. It is in the understanding that there is a need for action on both supply-side (service providers) and demand-side (patients/external clients) that the client service charter has to cover both the rights and responsibilities of patients as well as rights and responsibilities of health workers [16].

In Kenya, several challenges have been reported that affect the use of client service charter which include health workers not adhering to the charter, lack of time on the side of community members to read and understand the charter, and sociocultural issues [17].

Assessment of utilization of the client service charter launched by the Ministry of Health in Tanzania in 2005, found that most respondents had not heard of it, indicating that its distribution was inadequate, due to production of inadequate copies, lack of implementation strategy, advocacy and monitoring plan [18]. With the efforts put in place after the assessment to ensure its coverage, this paper aimed to use the available star rating dataset to establish extent of availability and implementation of the client service charter in PHC facilities (dispensaries, health centres and level one hospitals) both for public and private and determine if its availability and implementation is associated with client's satisfaction. PHC facilities account for more than 95% of all Tanzanian healthcare facilities. These are widely distributed in each and every 26 regions and 184 district councils of the country. There are about 60 million population in the country with equal access to services from PHC facilities regardless of gender, or ethnicity [19].

The results of this paper will contribute in filling some of the evidence gaps in PHC policy and governance, in particular gaps in–"*interventions to improve accountability for better governance in PHC; interventions to ensure transparency in local level decision making and governance; and role of user–provider communication in PHC to increase awareness and demand from user end which ensure better service and governance*" [20].

The study's main objective was to determine clients' satisfaction at PHC facilities and its association with the status of CSC implementation in Tanzania. Specifically, the objectives were to determine the: status of CSC implementation in Tanzanian PHC facilities using the SRA dataset of 2017/18; the proportion of PHC facilities whose clients were satisfied with services provided during a day of assessment; and predictors of clients' satisfaction at Tanzanian PHC facilities.

**Table 1. Service area 7—client focus.**

| 7 | Client Focus | Indicator |
|---|---|---|
| | 7.1. Client services charter | Client services charter displayed |
| | | Client services charter is monitored |
| | | Client feedback mechanism and complaints handling |
| | 7.2. Client satisfaction | Clients satisfied with services provided |

## Methods

### Conceptual framework

Implementation of policy guidelines such as client service charter require managers to be aware of organizational context (culture and trust), relationship management and negotiation of values in order to get support of health workers [21]. Health workers' non-adherence to the provisions of client service charter has been found to affect client's use of the charter, which means that there is a need of a strong mechanism for monitoring its implementation [16, 17]. Also, display of client service charter in health facilities alone is not enough, it requires to be innovative in coming up with ways that will raise awareness of clients on their rights and responsibilities depending on the local context [22]. In Tanga Region, an intervention in two hospitals, that consisted of implementation of the client service charter and a facility-based quality improvement process was found to have the potential for addressing disrespectful care during childbirth [23]. We conceptualized that availability of a client service charter that is displayed in various services delivery points, with a strong monitoring of its implementation, coupled with availability of a feedback and complaints handling system (all taking place in a context influenced by organizational culture and trust), will lead into improved implementation of a patient/client-centred care [24] leading to client satisfaction. Taking into account the difference between patient centered care and person-centered care in which the latter is broader capturing the "*goal of organizing care around the total (preventive and curative) needs and circumstances of each person, not merely around a disease category*" [15], we adopted the term person-centered care as intermediate outcome in the framework. The conceptual framework is shown in **Fig 1**.

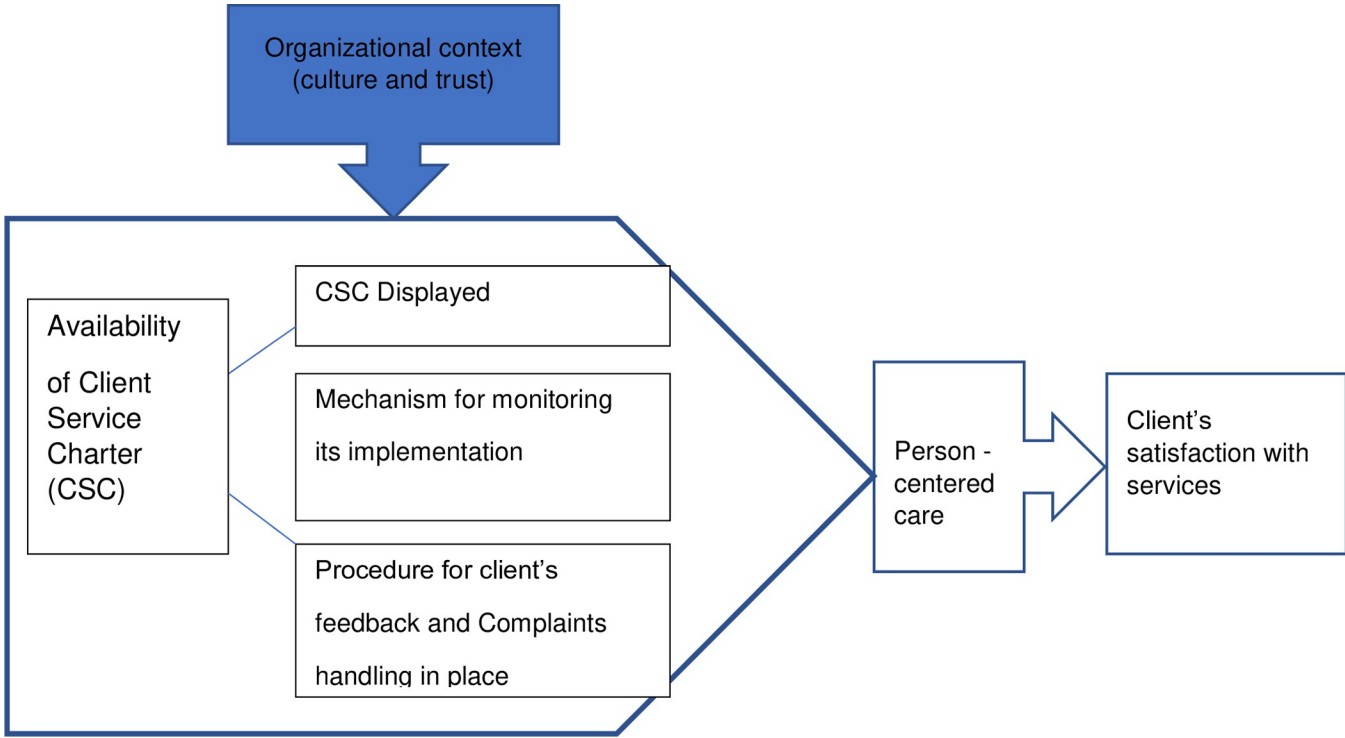

**Fig 1. Conceptual framework for implementation of client service charter in health facilities.**

## Study design

We extracted and performed a cross-sectional secondary data analysis of data related to clients' focus that are found in the national SRA database of 2017/2018.

## Study population

All PHC facilities registered in Tanzania were included in the SRA assessment of 2017/2018 and were the target for this study [4]. However, during data extraction, some had missing data and were excluded from the analysis.

PHC facilities are categorized in three levels in order of increasing capacity of healthcare service delivery; dispensary, health centre and hospital level 1 [25]. Dispensaries provide exclusively outpatients' services to approximately 10,000 population while health centres are referral points for dispensaries that provide a broader range of services including inpatient services, Comprehensive Emergency Obstetric, and Newborn Care (CEmONC) to about 50,000 population [25]. The hospital at level 1 serves about 250,000 population and receives referral from the low levels [25].

These PHC facilities are distributed widely in all 184 country's local government authorities shortly referred to as councils. Based on social-economic status, the councils are classified as to rural or urban councils [25]. Furthermore, PHC facilities are either publicly owned or privately owned. Publicly owned facilities include military facilities, police facilities, prisons facilities, parastatal facilities or council facilities. Private owned facilities are either non-government organizations, faith-based organization or private-for-profit organizations [4].

## Study variables and collection of data

**Data sources.** The data for this study was extracted from the DHIS2 database in Microsoft Excel Sheets format. The sheets were checked for data quality and cleaned before use. The database at the national level is managed by the Health Services Inspectorate and Quality Assurance Section of the then Health Quality Assurance Division, which is currently the Health Quality Assurance Unit (HQAU) of the Ministry of Health [13].

Data collection from each facility was done by at least four trained personnel; each from all healthcare administrative levels, i.e., national, regional, council and facility-level to ensure transparency and fairness [4].

**Dependent variable.** Client satisfaction was the main dependent variable of interest for this study. To determine whether a client was satisfied with their visit at the facility; a structured exit interview of clients selected from various service points was conducted. Three clients were randomly selected at each dispensary and five at health centres and hospital level 1. If there were more than one client exiting the facility, the first client to be interviewed was randomly selected by using a lottery system. In this system, each client was assigned a number and thereafter each number was written on each piece of paper. All papers were shuffled well and one paper was picked randomly to obtain the first interviewee. After returning from an interview, the data collector selected the next patient entering the consultation room until the targeted sample size was achieved. This way of conducting exit interviews is regarded by scholars as less biased and most operationally efficient [26]. The 10-point exit interview was used to score client satisfaction. A facility was considered satisfying clients if a reported average satisfaction score for all clients interviewed was at least 8 out of 10. This cut-off point is provided in the National Guidelines for Recognition of Implementation Status of Quality Improvement Initiatives in Health Facilities [27]. The content of the structured exit interview is shown in **Table 2.**

**Table 2. An exit interview tool used during the Star Rating Assessment of 2017/18 in Tanzania.**

| | CLIENT EXIT INTERVIEW: client over age 18 or representative if client under 18 Client number 1 / 2 / 3 / 4 / 5 | | |
|---|---|---|---|
| | Explain the purpose of this 5 minute interview to the client (or the client's representative in the case of minors) and obtain consent. Purpose: "We are here to give a star rating for this facility. May I ask you a few questions to know if your visit was satisfactory?" | | |
| | Interviewer code \|___\|___\|___\|___\| Time (HH:MM) \|___\|___\|: \|___\|___\| | | |
| | Client age | Years \|___\|___\| | |
| | Client sex | Male / Female | |
| 1 | What time did you arrive at the facility? How long did you wait before you were seen by a health worker? (total wait from arrival time until time seen by health worker), | Time \|___\|___\|: \|___\|___\| Minutes \|___\|___\| Wait <60 minutes? Yes / No | HH:MM Wait less than 60 minutes Yes = 1 point |
| 2 | Was the waiting time acceptable to you? | Yes / No | Yes = 1 point |
| 3 | Did the health worker examine you? | Yes / No | Yes = 1 point |
| 4 | Did the health worker explain about your care, or illness, and about any tests or treatment? | Yes / No | Yes = 1 point |
| 5 | Did you receive all the prescribed medicines? | Yes / No | Yes = 1 point |
| 6 | Did you understand how to take the medicines? [probe] | Yes / No | Yes = 1 point |
| 7 | Were the health workers polite and respectful? | Yes / No | Yes = 1 point |
| 8 | Did you have enough privacy during your visit? | Yes / No | Yes = 1 point |
| 9 | Did you find the facilities clean and in order? | Yes / No | Yes = 1 point |
| 10 | Are the fees and charges fair and affordable to you? [Q also applies to CHF/ NHIF members] | Yes / No | Yes = 1 point |
| | | SCORE \|___\|___\| points | (out of 10 points) |
| | **Was your visit satisfactory overall?** If not satisfactory, please explain: | **Yes / No** | Not scored. Record comments |
| | What would you like to see improved? | | Record comments |
| | If not all medicines received, what do you do? | | Record comments |
| | Thank the person for their participation. | | |

**Independent variables.** Three indicators of client service charter namely display of client services charter, monitoring of client charter and presence of client feedback mechanism and complaints handling were considered independent variables to client satisfaction. The indicator for the client service charter was regarded implemented only if all verification questions scored "Yes "for that particular indicator The detailed assessment criteria and scoring scale of these indicators is presented in **Table 3**.

Other independent variables were facility's characteristics such as location (rural or urban), health facility level (dispensary, health centre or hospital level 1) and health facility ownership (public or private).

## Analysis

The data were checked for completeness manually, entered into Epi-info version 7.2.2.6 and then transferred into STATA version 15 for analysis. Status of client service charter implementation was presented by after calculating the proportion of PHC facilities that scored "yes" in each of the three indicators. The clients were satisfied in the facilities that scored "yes". This was presented as the proportion of facilities whose at least 80% of their clients were satisfied

**Table 3. Assessment criteria and scoring scale for client service charter indicators during SRA 2017/28.**

| NO. | INDICATOR | QUESTION & VERIFICATION METHOD | RESPONSES (Y = yes; P = partial; N = no) |
|---|---|---|---|
| 7.1 | **Client Service Charter** | | |
| 7.1.1 | **Client service charter displayed** | 1. Is the client service charter available at this facility? | Y. Client charter available<br>N. Client charter not available |
| | | 2. Is the client service charter displayed in a public area, and visible to clients | Y. Client charter well displayed<br>N. Client charter not well displayed |
| 7.1.2 | **Client service charter is monitored** | 1. Does the facility management team measure compliance with the client service charter?<br>*Check if documented within the last six months* | Y. Compliance with client charter is assessed and measured by the facility management team<br>N. No assessment or measurement of compliance with the client charter |
| 7.1.3 | **Client feedback mechanism and complaint handling** | 1. Is any method for client feedback in place at the facility?<br>*Any of the following will qualify: suggestion box, client help desk, display of contact details for phone or SMS feedback. Specify the method(s) in use.* | Y. Feedback method in place<br>N. No feedback method place<br>If 'Y' specify the methods in place:<br>______________________________ |
| | | 2. Is the feedback mechanism in use?<br>*Check records of complaints/ suggestions over the last 6 months* | Y. Records indicate a feedback mechanism is used<br>N. No records in the last 6 months, feedback mechanism not working |
| | | 3. Has there been any action on suggestions for improvement, or to address complaints from the feedback mechanism?<br>*Check documentation on actions and any improvement.* | Y. There is a record of actions to take up suggestions/ address the complaints<br>N: There is no record of action or no feedback mechanism |
| | | 4. Is there any community participation and engagement arising from the feedback mechanism?<br>*Check if WEO, HFGC members, or CHW are present for opening of the suggestion box, or information provided to HFGC or community from feedback mechanism.* | Y. Community engaged or information shared<br>N: No engagement or sharing of information |

with service delivery on assessment day. Based on this cut point, a binary variable was created with two values; yes-for the facilities whose clients are satisfied and no-for those whose clients were not satisfied. The binary variable was used to determine the predictors of the clients' satisfaction that was used to determine an association between the facilities' client satisfaction and independent variables.

GIS software version 2.8.6 was used to display the spatial distribution of the Proportion of Tanzanian Primary Health Facilities whose clients were satisfied with services. The shapes file that was used to construct the map is unrestrictedly available at the National Bureau of Statistic's website: https://www.nbs.go.tz/index.php/en/census-surveys/gis

### Ethics statement

The SRA was conducted under ethical approval granted independently from this study. This study did not require additional approval as it was a secondary analysis of anonymised data.

## Results

### Description of health facilities under study

By 2017 Tanzania had 7,289 PHC facilities that were all involved in SRA assessment of year 2017/2018. A total of 4,523 of the assessed facilities met our inclusion criteria and therefore used for final analyses. These included 3,987 (88.2%) dispensaries, 408 (9.0%) health centres and 128 (2.8%) hospitals. 952 (21%) of facilities were located at urban-settings while 3,571 (79%) were found at rural. Out of 4,523; 3,714 (82.1%) and 809 (17.9%) were public-owned and private-owned respectively. A separate analysis was done and found no significant difference among excluded facilities and those involved in this study (Table 4).

**Table 4. A separate analysis output for 2766 (38%) PHC facilities that were excluded from final analysis.**

| Variable | %Facilities involved in data collection | %Facilities included in the analysis | p-value* |
|---|---|---|---|
| **Facility type** | | | |
| Health Centre | 3.0 | 2.8 | 1.0 |
| Hospital | 12.0 | 9.0 | |
| Dispensary | 85.0 | 88.2 | |
| **Ownership** | | | |
| Private | 19.0 | 17.9 | 1.0 |
| Public | 81.0 | 82.1 | |
| **Location** | | | |
| Urban | 22.4 | 21.0 | 1.0 |
| Rural | 77.6 | 79.0 | |

*p-value was calculated from paired t-test

## Status of client's service charter implementation in PHC facilities

The clients' service charter was displayed in 69.1% facilities, monitored in 32.4% facilities, and 32.5% of the facilities had mechanisms for clients' feedback and handling complaints (**Fig 2**).

## Clients satisfied with services provided at PHC facilities

About seventy-two (72.2%) of PHC facilities had their clients satisfied with services provided on a day of assessment as shown in **Fig 2**. Prevalence of satisfaction among public-owned

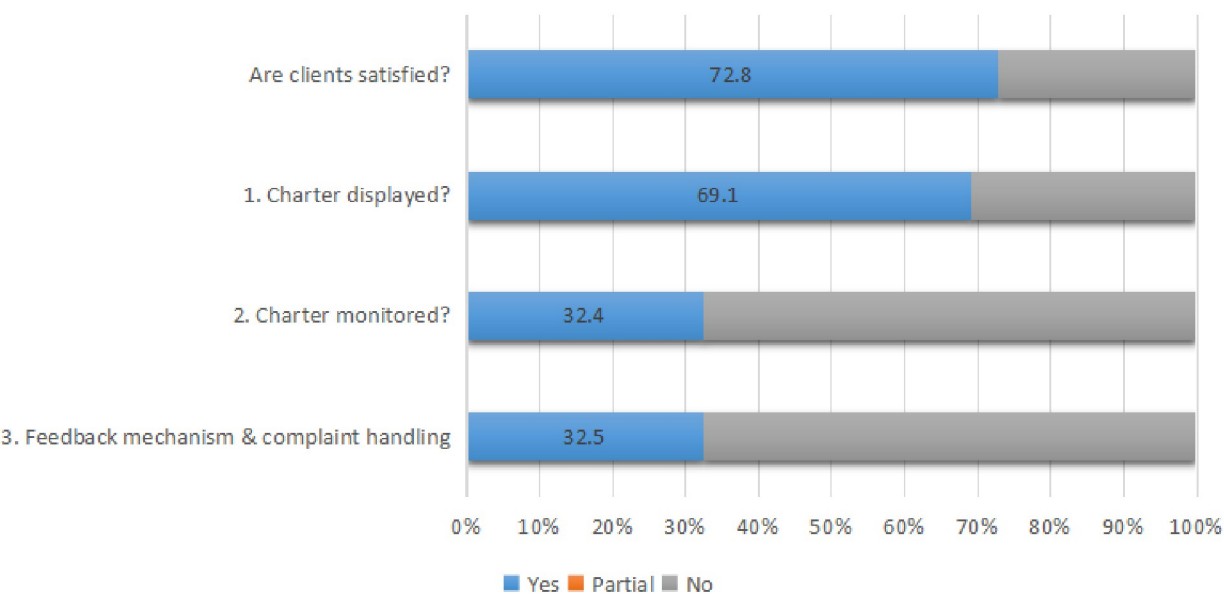

**Fig 2. Proportion of primary healthcare facilities whose clients are satisfied with services and the status of implementation of clients' service charter in Tanzania.**

**Table 5. Predictors of clients' satisfaction at primary healthcare facilities during SRA of 2017/18.**

| Variable | Clients satisfaction | | | | Bivariate | | | Multivariate | | |
|---|---|---|---|---|---|---|---|---|---|---|
| | Yes | % | No | % | COR | 95% CI | p-value | AOR | 95% CI | p-value |
| **Facility type** | | | | | | | | | | |
| Health Centre | 282 | 69.1 | 126 | 30.9 | 0.83 | 0.55–1.25 | 0.373 | 0.94 | 0.58–1.53 | 0.817 |
| Hospital | 98 | 76.6 | 30 | 23.4 | 1.21 | 0.97–1.51 | 0.093 | 1.39 | 1.10–1.77 | 0.006 |
| Dispensary | 2,911 | 73.0 | 1,076 | 27.0 | Ref | | | Ref | | |
| **Ownership** | | | | | | | | | | |
| Private | 603 | 74.5 | 206 | 25.5 | 1.12 | 0.94–1.33 | 0.211 | 1.19 | 0.97–1.46 | 0.095 |
| Public | 2,688 | 72.4 | 1,026 | 27.6 | Ref | | | Ref | | |
| **Location** | | | | | | | | | | |
| Urban | 710 | 74.6 | 242 | 25.4 | 1.13 | 0.96–1.32 | 0.156 | 1.21 | 1.00–1.46 | 0.043 |
| Rural | 2,581 | 72.3 | 990 | 27.7 | Ref | | | Ref | | |
| **Charter displayed?** | | | | | | | | | | |
| Yes | 2,368 | 78.4 | 651 | 21.6 | 1.91 | 1.66–2.20 | <0.001 | 1.62 | 1.39–1.89 | <0.001 |
| No | 885 | 65.6 | 465 | 34.6 | Ref | | | | | |
| **Charter monitored?** | | | | | | | | | | |
| Yes | 1,150 | 82.3 | 248 | 17.7 | 1.92 | 1.64–2.25 | <0.001 | 1.47 | 1.23–1.75 | <0.001 |
| No | 2,059 | 70.7 | 852 | 29.3 | Ref | | | | | |
| **Feedback mechanism &complaint handling** | | | | | | | | | | |
| Yes | 1,120 | 77.2 | 330 | 22.8 | 1.35 | 1.16–1.56 | <0.001 | 1.45 | 1.23–1.72 | <0.001 |
| No | 2,154 | 71.6 | 855 | 28.4 | Ref | | | | | |

*p*- Values are calculated using chi square test

*Predictors whose association were found significant in the final logistic regression model

COR = Crude/unadjusted Odds Ratio, AOR = Adjusted Odds Ratio, Ref = Reference group

facilities was 72.4% and 74.5% in private-owned facilities. Overall, clients from hospitals had the highest prevalence of satisfaction followed by dispensaries and health centres (76.6% versus 73.0% versus 69.1%, respectively).

## Predictors of clients' satisfaction at PHC facilities

**Table 5** shows the results of bivariate and multivariate logistic regressions with client's satisfaction at PHC as the dependent variable and PHC characteristics and performance indicators of client's service charter as the predictor variables. With reference to dispensaries, clients who received services in hospitals were more likely to be satisfied with services (AOR 1.39; 95% CI: 1.10–1.77) but not clients from health centres (AOR 0.94; 95% CI: 0.58–1.53). Those who received services from urban-based PHC facilities were more likely to be satisfied compared to those received services from rural-based facilities (AOR 1.21; 95% CI: 1.00–1.46).

There was no significant difference in clients' satisfaction between clients who received services in private facilities and in public facilities. All indicators of client service charter's performance (i.e., display of charter, charter monitoring and presence of client feedback mechanism and complaints handling) were significantly associated to clients' satisfaction.

## Discussion

### Status of clients' service charter implementation in PHC facilities

A client service charter is a *document that sets out information on the services provided, the standards of service that customers can expect from the facility, and how to make complaints or*

*suggestions for improvement"* [28, 29]. Therefore, improved implementation of a patient/client-centred healthcare relies on display of the charter in various services delivery points, effective charter monitoring, and availability of a feedback and complaints handling system [24].

Findings of this study show the charter was displayed at slightly above two-thirds of the facilities (69.1%), nevertheless monitored in about one-third of these facilities (32.4%). This means two-thirds of the facilities understudy never implemented the charter fully. The findings are congruent with those found in a nearby country, Uganda whereby the charters were displayed on the walls of the facilities; yet, implementation of the charter was not done because patients were not aware of their rights [22]. Contrary to our findings, studies from other low and middle-income countries have indicated the challenge of adoption of the service charter whereby less than half of the facilities had charter displayed [30, 31]. In Tanzania for example, findings before the conduct of this study had concluded that client service charter in Tanzania was not a reality rather a myth [32]. We presume the situation improved due to impact of implementing SRA system initiatives since 2015 that (among others) focus on improving implementation of client service charter in PHC facilities [4, 33].

Implementation of the charter is strangled by lack of motivation among healthcare workers, interference from political leaders, culture, and inadequate work force and finances [34–36]. According to Thomassen et al. [29], a good performance in client charter requires that employees are motivated and stimulated over the charter. Findings from Tanzania [37] and other neighbouring countries Kenya [35] and Uganda [36] suggest staff in medical field are less psychologically and economically motivated to implement the charter and therefore motivation should target the economic and psychological needs of this group. Psychological motivation may be achieved through regular workshop and trips, while economy and morale could be raised through staff appraisal and promotions [35, 38]. In Tanzania, healthcare workers could even take ten years in service without any promotion [38]; and some of them believe that clients satisfaction comes after employees satisfaction [39]. Despites financial constraints that lead to motivational challenges in low and middle income countries; we suggest the use of alternative but more effective bottom-top strategies of motivation such as rewarding best performers at facility level and improved working environment [40].

Our findings reveal that there was neither mechanisms for obtaining feedback from the clients nor mechanisms for handling the complaints should they arise in majority of the facilities. The facility was considered using the feedback mechanisms if there were records of client's complaints lodged through either suggestion box, client help desk, phone or short text message (SMS) over the period of past six months before the day of an assessment. According to our assessment criteria, facilities were required to either invite community members during opening of suggestion boxes or share to community leaders the clients' complaints obtained through other methods than suggestion boxes. The involvement of opinion leaders in implementing the charter was expected to bring in the power which could be used to achieve the goals and content of the charter by employees [29]. However, community participation in Tanzanian health system has been politicized and influenced by leaders who are inflexible, tardy, non-cooperative and corrupt [34–36]. These attributes weaken community engagement that leads to poor management of the client service charter. Community participation could be improved through effective preparation of the meetings, providing community representatives with timely feedback and set the funds needed to facilitate such gathering and improved working environment [41].

Healthcare workers have a big role to play in effective implementation of the charter. Healthcare workers are reported to lack knowledge and patient-centred culture needed to implement the client service charter [29]. Orientation of health workers on client's service charter and strengthening complaints management system needs would improve implementation of client service charter in primary health facilities in Tanzania [42].

## Client's satisfaction at PHC facilities and the associated predictors

Satisfaction among clients who visited the facilities was as high as 72.8%, nevertheless below the recommended minimum value of 80%. The prevalence reported in our study could be considered among the higher prevalence that has ever been reported in low and middle-income countries [43–46]. The high satisfaction scores in Tanzania could be attributed to SRA initiatives that commenced four years prior to the conduct of this study. These initiatives include instruction to facilities on developing and implementing individual client service charters. The impact of the initiatives was to be revealed in the next assessment of which its findings are presented in this study. Previous studies that were done before SRA showed lower satisfaction [43–46].

All three indicators of charter performance (i.e., display of client services charter, monitoring of client charter, and presence of client feedback mechanism and complaints handling) were significant predictors of clients' satisfaction. This means improving any of the indicators of the client service charter is likely to result in increased clients' satisfaction. Implementation of the charter's indicators increases commitment among healthcare providers to providing quality services and therefore increases clients' satisfaction [43–46].

Unexpectedly, we found no significant difference in clients' satisfaction among clients that obtained services from private and those from public owned facilities. Other studies have reported higher satisfaction among clients who attended at private facilities compared to those from public-owned facilities [43, 47]. Higher satisfaction within private facilities has been linked with availability of equipment, materials, infrastructures, commodities and motivated staff [48] and thus readiness for customer service compared to public facilities [49]. In contrary to the above, there is an emerging evidence that clients from developing countries such as Tanzania are cost-conscious and therefore are likely to be satisfied from facilities that cost services at cheaper prices [43–46].

Our findings show clients were likely to be satisfied from hospital-level services but not from health centre services, considering dispensaries as the reference. In Tanzania, health centres unlike dispensaries; are intended to provide addition of inpatient services plus Comprehensive Emergency Obstetric and Newborn Care services (CEmONC) [25]. However, majority of the health centres operate basically as dispensaries since they do not provide CEmONC services [45, 50–52]. Provision of safe deliveries in Tanzania, the main objective of CEmONC, has been hugely associated with clients' satisfaction [45, 50–52]; therefore, we can associate low client satisfaction at health centres with inadequate quality services. In nutshell, we learned that clients satisfaction is more related to the services provision rather than facility level [44, 47].

Clients were likely to be satisfied from facilities based in urban settings compared to those based in rural areas. The findings are in agreement with those from other low and middle income countries in which urban-based facilities were likely to be more customer-focused and equipped with more equipment and commodities [49], human and financial resources [52, 53], and service bundles [54] compared to rural-based facilities.

Ecological zone is a known predictor of client satisfaction [55]. Clients from different zones could have varied education level and expectation from the services provided at health facilities and hence different in level of satisfaction [56]. **Fig 3** shows regions scored more or less equally in proportion of facilities that had clients satisfied from service provision. Few regions that on average achieved the recommended scores in clients' satisfaction (labelled in green) came from different ecological zones that had no equal pattern of either socio-economic status or activities.

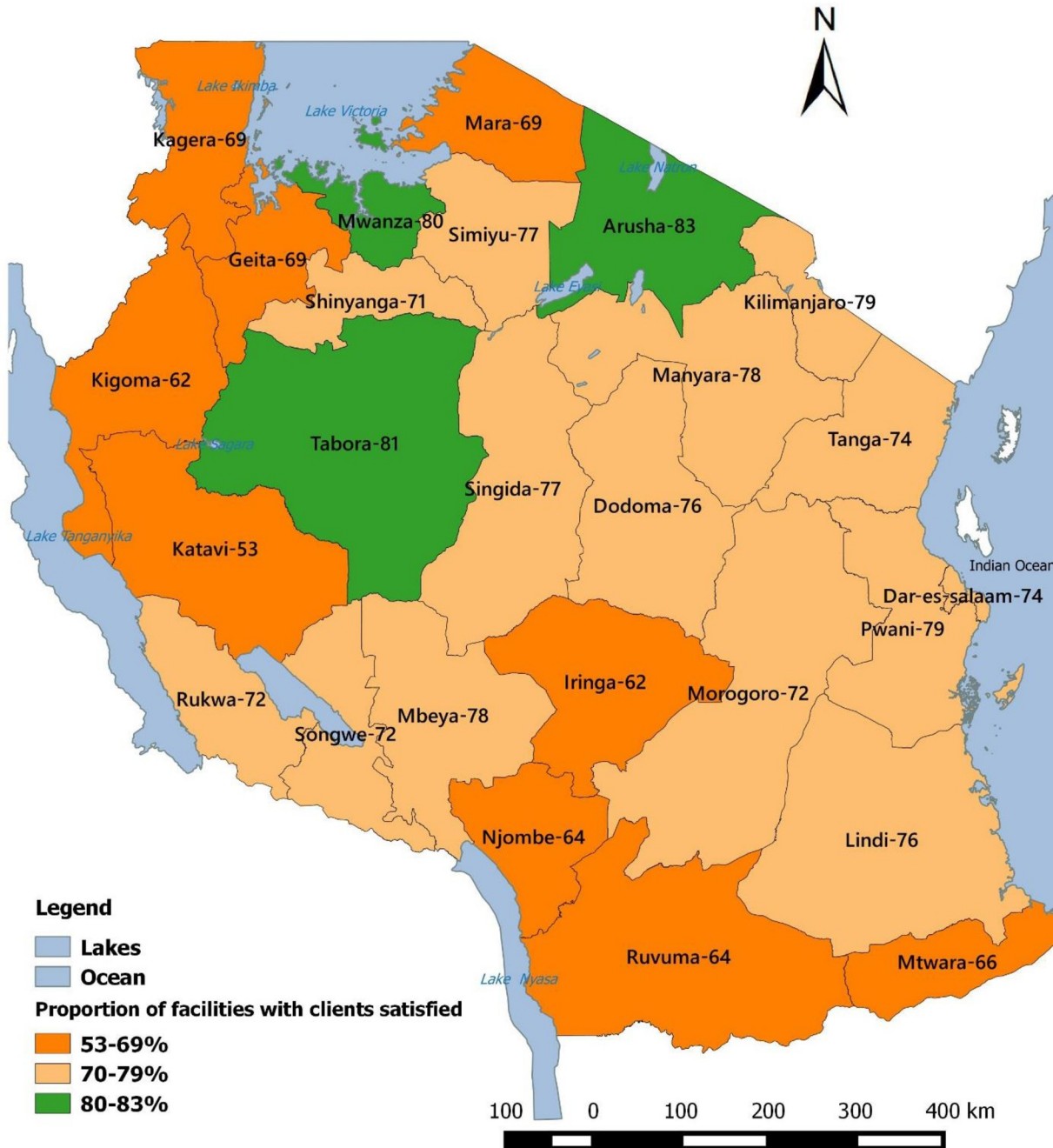

**Fig 3. Proportion of Tanzanian primary health facilities whose clients were satisfied with services during star rating assessment 2017/118.**

### The implication of the study

This paper has three policy implications. First, the Ministry published a "National Client's Service Charter for Health Facilities" in 2018 aiming at providing guidance to health facilities on how to adapt the charter for use at all areas of service delivery. The results have provided a baseline and lessons for the Ministry to closely assess, monitor and evaluate implementation of client's service charter in health facilities [16]. Second, the coronavirus disease of 2019 (COVID-19) has exacerbated the already weak health systems in low- and middle- income

countries threatening to disrupt services [57]. The COVID-19 poses a direct risk to health facilities compliance to client service charter. Therefore, these finding will help the Ministry and all stakeholders to further collaborate in strengthening PHC facilities to implement their client service charter in order to improve the quality of health care services provided. Third, investing in strengthening implementation of client service charter will help to improve the PHC facilities responsiveness to client's needs and expectations and hence, improving person-centred care. A study conducted in four low- and middle- income countries (LMICs) has shown that improvements in person-centred care are essential for a high quality health system [58].

## Strengths and limitations

This is the first Tanzanian study on client focus assessment having National coverage of PHC facilities. The findings will allow fair comparisons with similar studies elsewhere thus informing policy makers and health planners globally.

Our dataset had limited information on patients' characteristics therefore we were unable to determine if these characteristics were associated to clients' satisfaction. Nevertheless, previous studies have shown no significant association between socio-demographic predictors and client satisfaction [55].

Our findings that show relatively high reported clients satisfaction level should be translated with caution as the results could have been biased by the data collection method, i.e., exit interview. The '*white coat effect*' introduced by exit interviews could have exaggerated the values of the results since a client is likely to not say anything bad against the facility which he/she gets service from [59]. However, data collectors were instructed to conduct exit interviews outside the facilities' compound during data collection.

Finally, three to five may not be the right representative number of clients to conclude on clients' satisfaction at a given facility. However, the experience during data collection showed the number of clients attending at the majority of the facilities i.e. dispensaries was very low to the extent that it would be impossible to get an extra interviewee at some facilities. The findings of this study are presented as "satisfaction on a day of the visit" and should be translated with such a precaution.

## Conclusion and recommendations

The status of implementation of client service charter is very low among Tanzanian PHC facilities. Majority facilities had the charter displayed at facilities but not monitored. Furthermore, the proportion of PHC facilities whose clients were satisfied with services provided during a day of visit is very convincing though falls below the recommended value of 80%. Clients were likely to be satisfied if received healthcare services from facilities that display the charter, monitor the implementation of the charter and from the facilities that have mechanisms to obtain clients feedback and handle complaints that arise from service delivery.

Moreover, clients had higher satisfaction if were served from the facilities that are based in urban areas and by highest level PHC facilities included in this study, i.e., hospital-level. According to Tanzanian context; these kind of PHC facilities are likely to offer better services so do higher clients satisfaction. Therefore, more effort in improving quality of services should target health centres and PHC facilities that are based in rural areas.

The implementation of the client service charter among facilities is low compared to how clients are satisfied. We assume there is a huge improvement in the clients experience towards healthcare delivery since inception of the SRA initiatives.

The client service charter is an approved tool for managing performance and quality of service delivery in Tanzania [32]. Poor performance in client service indicates that PHC facilities were providing services below the expected level of patient-centeredness and focus and therefore more efforts is needed to improve the situation.

## Recommendations

In order to improve the implementation of client service charters in primary health facilities in Tanzania, the orientation of health workers on client's service charter and strengthening complaints management system needs to be taken into account [42]. It is also important for the Ministry to embed implementation research in implementation of the SRAs in PHC facilities in order to learn along the process and hence develop a learning health system in Tanzania [60].

## Acknowledgments

The authors are passing their sincere gratitude to the Ministry of Health, Community Development, Gender, Elderly and Children (MoHCDGEC) especially, the Health Quality Assurance Unit for granting us permission to use the SRA data.

Apart from government institutions, the authors extend appreciation to development partners such as World Bank, The United States—Centres for Disease Control and Prevention (CDC), Danish International Development Agency (DANIDA), and The World Health Organization whom together made SRA possible. Others were the communities of the facilities visited, PharmAccess International, Association of Private Health Facilities in Tanzania (APHTA), Christian Social Services Commission (CSSC), and Development Partners in Health-Group (DPG-H).

## Author Contributions

**Conceptualization:** Erick S. Kinyenje, Talhiya A. Yahya, Mbwana M. Degeh, Chrisogone C. German, Joseph C. Hokororo, Mohamed A. Mohamed, Omary A. Nassoro, Radenta P. Bahegwa, Yohanes S. Msigwa, Ruth R. Ngowi, Laura E. Marandu, Syabo M. Mwaisengela, Eliudi S. Eliakimu.

**Data curation:** Erick S. Kinyenje, Talhiya A. Yahya, Chrisogone C. German, Joseph C. Hokororo, Ruth R. Ngowi, Eliudi S. Eliakimu.

**Formal analysis:** Erick S. Kinyenje.

**Methodology:** Erick S. Kinyenje, Mbwana M. Degeh, Joseph C. Hokororo, Eliudi S. Eliakimu.

**Software:** Erick S. Kinyenje.

**Supervision:** Talhiya A. Yahya, Mbwana M. Degeh, Chrisogone C. German, Eliudi S. Eliakimu.

**Validation:** Erick S. Kinyenje, Joseph C. Hokororo, Mohamed A. Mohamed, Radenta P. Bahegwa, Ruth R. Ngowi, Eliudi S. Eliakimu.

**Visualization:** Erick S. Kinyenje, Ruth R. Ngowi, Eliudi S. Eliakimu.

**Writing – original draft:** Erick S. Kinyenje, Mbwana M. Degeh, Chrisogone C. German, Joseph C. Hokororo, Omary A. Nassoro, Radenta P. Bahegwa, Yohanes S. Msigwa, Ruth R. Ngowi, Laura E. Marandu, Syabo M. Mwaisengela, Eliudi S. Eliakimu.

**Writing – review & editing:** Erick S. Kinyenje, Talhiya A. Yahya, Mbwana M. Degeh, Chrisogone C. German, Joseph C. Hokororo, Mohamed A. Mohamed, Omary A. Nassoro, Radenta P. Bahegwa, Yohanes S. Msigwa, Ruth R. Ngowi, Laura E. Marandu, Syabo M. Mwaisengela, Eliudi S. Eliakimu.

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
