## [Decision Letter · Decision Letter 0]

26 Jan 2022

PONE-D-21-39135Clients satisfaction at primary health-care facilities and its association to status of client service charter implementation in TanzaniaPLOS ONE

Dear Dr. Erick Kinyenje,

Thank you for submitting your manuscript to PLOS ONE. After careful consideration, we feel that it has merit but does not fully meet PLOS ONE’s publication criteria as it currently stands. Therefore, we invite you to submit a revised version of the manuscript that addresses the points raised during the review process.

In this review, the opposite decision was made between reviewers and, thus, more reviewers were included. There are several major concerns that need further revision and I believe that this work will be more mature during the revision.

We look forward to receiving your revised manuscript.

Kind regards,

Wen-Wei Sung, M.D., Ph.D.

Academic Editor

PLOS ONE

[I have read the journal's policy and the authors of this manuscript have the following competing interests: During the time of baseline and reassessment as well as during write up of the study – JH, EE and TY were with the Health Quality Assurance Divison (now called Health Quality Assurance Unit) and were responsible for the implementation of SRA and QIPs folllow-up.] 

3. We note that Figure 3 in your submission contain map images which may be copyrighted. All PLOS content is published under the Creative Commons Attribution License (CC BY 4.0), which means that the manuscript, images, and Supporting Information files will be freely available online, and any third party is permitted to access, download, copy, distribute, and use these materials in any way, even commercially, with proper attribution. For these reasons, we cannot publish previously copyrighted maps or satellite images created using proprietary data, such as Google software (Google Maps, Street View, and Earth). For more information, see our copyright guidelines: http://journals.plos.org/plosone/s/licenses-and-copyright.

a) You may seek permission from the original copyright holder of Figure 3 to publish the content specifically under the CC BY 4.0 license.  

Reviewers' comments:

Reviewer's Responses to Questions

**Comments to the Author**

1. Is the manuscript technically sound, and do the data support the conclusions?

Reviewer #1: Yes

Reviewer #2: No

Reviewer #3: Yes

Reviewer #4: No

Reviewer #5: Yes

2. Has the statistical analysis been performed appropriately and rigorously? 

Reviewer #1: Yes

Reviewer #2: No

Reviewer #3: Yes

Reviewer #4: No

Reviewer #5: Yes

3. Have the authors made all data underlying the findings in their manuscript fully available?

Reviewer #1: Yes

Reviewer #2: No

Reviewer #3: Yes

Reviewer #4: Yes

Reviewer #5: Yes

4. Is the manuscript presented in an intelligible fashion and written in standard English?

Reviewer #1: No

Reviewer #2: No

Reviewer #3: Yes

Reviewer #4: Yes

Reviewer #5: Yes

5. Review Comments to the Author

Reviewer #1: Manuscript should be revised by a native English speaker. Can you tell some reasons for facilities not being available in the data base? How was random selection of persons to interview organized? List of references could be shortened.

Thanks for your efforts!

Reviewer #2: DEAR Author,

The study’s topic, “Clients satisfaction at primary healthcare facilities and its association to the status of client service charter implementation in Tanzania” it’s interesting, but some sections should be clarified.

-The methodology is unclear, such measure tool or scare lack references support, and lack reliability and validity, and “Explain the purpose of this 5-minute interview to the client…” how to mention the Questionnaire could be measured.

-In addition, the study’s satisfaction score points present dichotomy, only yes or no, why result could be presented percentage.

-Result section, Figure 2: the satisfaction with services data why lower more than 80%, suggest author address more.

-“Table 4. Predictors of clients’ satisfaction at healthcare facilities during SRA of 2017/18.” incorrectly presented, samples could not match the study’s text, please check.

-For more contributions, suggest authors should clarify each clients satisfaction item and analyze the samples which clients satisfaction item is more meaningful to be improved. Such as patient characteristics or facilities health workers’ quality care.

-The discussion section should be reworded based on the incorrect analysis result.

-Page 18, “Second, the coronavirus disease of 2019 (COVID-19) has exacerbated the already weak health systems in low- and middle- income countries threatening to disrupt services…” it does not include the study range, so I am confused because page 11 authors presented “….By 2017 Tanzania had 7,289 PHC facilities that were all involved in SRA assessment of the year 2017/2018.”

Thank you for your effort.

Reviewer #3: Introduction

Those who are not familiar with primary healthcare do not really know what the Astana Declaration is. Please elaborate a bit more in detail.

Patient satisfaction in primary care is a controversial outcome as international studies have shown ceiling effects. Please discuss.

On page 4 it is stated „This paper discusses service area number seven (7) of the star rating assessment (SRA) tools, namely client focus.“ without explaining the SRA tools. Therefore, the reader is not able to follow the context.

It is not clear why the SRA dataset of 2017/18 was used. Why was there no newer data?

Methods

In the Conceptual Framework the details of the content and implementation of the client service charter should be described. Is there a SOP which describes how to implement the CSC? How is it measured if a CSC is implemented or not?

„due to one reason or another, facilities whose results were not found in the database were excluded from the analysis.“ => These reasons have to be explained in detail. Otherwise you have a sample with unknown selection criteria which influence your results.

Table 3: „Client service charter was implemented if all three indicators were met.“ Does this mean that all the displayed questions in Table 3 had to be answered with „Yes“ ? Please explain.

The Analysis section has to be much more detailed. Statistical measures used have to be justified and described. The binary variable regarding patient satisfaction should be explained more clearly. Those who scored 8/10 were considered to be satisfied with the service?

Results

There are 2766 (38%) PHC facilities not included in the analyses. A missing analysis regarding their characteristics has to be performed in order to demonstrate if you have a selective sample or not.

Table 4:

As far as I have understood the authors have performed single-variable regressions first with each of the listed predictors and then included them all in one multivariate analyses. Please explain more precisely. Please give an indicator of model performance, e.g. McFadden´s R-square.

Discussion

Important points were raised for the future of the PHC system in Tanzania. Please explain further the discrepancy between the relatively low implementation of client service charter and the relatively high satisfaction of patients.

Reviewer #4: The logical comments are as below.

1. The authors defined "Three clients were randomly selected at each dispensary and five at health centers and hospital

level 1" as the facilities clients was satisfied with, and potential factors were examined for the associations with the client-satisfying facility. Three or five clients are not enough to evaluate the facilities in terms of client satisfaction. The authors should analyze the associations between client satisfaction and factors, not between facilities evaluated by a limited number of clients and factors.

2. The authors did not describe the method of random selection of clients. The readers cannot judge whether the clients were sampled randomly or not.

3. The characteristics of clients were not described. The readers cannot understand whose evaluation was the base of satisfaction.

4. The characteristics and qualification of interviewers were not described.

5. Figure 2 is hard to be interpreted. The title says "Proportion of primary health facilities whose clients are satisfied". For example, what did the author mean for the proportion of "Charter displayed?"?

The technical comments are as follows.

1. The abbreviation must be defined at the first appearance with the full spelling, and the abbreviation should be used thereafter. For example, "CSC".

2. In Table 2, delete "=" after "Wait less than 60 minutes".

3. In Table 4, COR of 1.125 should be replaced with "1.13". The term "Ref" is dropped at four parts.

Reviewer #5: Dear authors

I would like to thank you for giving me the opportunity to review this well-written manuscript entitled “Clients satisfaction at primary health-care facilities and its association to status of client service charter implementation in Tanzania”. The main objective of the study was to determine clients’ satisfaction at PHC facilities and its association with status of client service charter implementation in Tanzania. This work is well designed and well written manuscript that can improve the delivering PHC services. Therefore, I think the manuscript can be considered seriously for publishing. I have just one comment:

1. I would like to see some information regarding Tanzania such as population, socio-economic status, and PHC in the introduction.

Best wishes

6. PLOS authors have the option to publish the peer review history of their article (what does this mean?). If published, this will include your full peer review and any attached files.

Reviewer #1: No

Reviewer #2: No

Reviewer #3: No

Reviewer #4: **Yes: **Nobuyuki Hamajima

Reviewer #5: **Yes: **Abbas Mardani

---

## [Author Response · Author response to Decision Letter 0]

29 Jun 2022

PONE-D-21-39135

Clients satisfaction at primary health-care facilities and its association to status of client service charter implementation in Tanzania

PLOS ONE

Dear editor, we humbly submit the responses to comments raised by the reviewers including you. Thank you again for giving us the opportunity. 

Additional Journal requirements:

As per guidelines, all headings for major sections in the manuscript have been increased in font size from 16 to 18. 

As per guidelines, all headings for sub-sections of major sections in the manuscript have been increased in font size from 14 to 16. 

As per guidelines, all headings for level 3 in the manuscript have been increased in font size from 12 to 14. 

An extension “.tif” has been added to each figure name.

Figures are now cited as “Fig 1”, “Fig 2” and “Fig 3” instead of “Figure 1”, “Figure 2” and “Figure 3” respectively.

Figures are now captioned as “Fig. 1”, “Fig. 2” and “Fig. 3” instead of “Figure 1”, “Figure 2” and “Figure 3” respectively.

Figure titles are now bolded but not italicized. 

All titles have been written in sentence case

[I have read the journal's policy and the authors of this manuscript have the following competing interests: During the time of baseline and reassessment as well as during write up of the study – JH, EE and TY were with the Health Quality Assurance Divison (now called Health Quality Assurance Unit) and were responsible for the implementation of SRA and QIPs folllow-up.] 

Thank you. The statement is included in the cover letter as instructed.

3. We note that Figure 3 in your submission contain map images which may be copyrighted. All PLOS content is published under the Creative Commons Attribution License (CC BY 4.0), which means that the manuscript, images, and Supporting Information files will be freely available online, and any third party is permitted to access, download, copy, distribute, and use these materials in any way, even commercially, with proper attribution. For these reasons, we cannot publish previously copyrighted maps or satellite images created using proprietary data, such as Google software (Google Maps, Street View, and Earth). For more information, see our copyright guidelines: http://journals.plos.org/plosone/s/licenses-and-copyright.

a) You may seek permission from the original copyright holder of Figure 3 to publish the content specifically under the CC BY 4.0 license. 

Figure 3 is our own construction and was not accessed from other sources than us, authors.

Reviewers' comments:

Reviewer's Responses to Questions

Reviewer #1: Manuscript should be revised by a native English speaker. Can you tell some reasons for facilities not being available in the data base? How was random selection of persons to interview organized? List of references could be shortened.

Thanks for your efforts!

It is not true that some facilities were not available in the database, the true statement should have been “some facilities had missing/incomplete data”. We targeted all facilities that were assessed through Star Rating Assessment (SRA). However , some had missing data during data extraction and were excluded from the analysis. The methodology section has added this statement to clarify the previous statement.

How was random selection of persons to interview organized?

We have added a description (in data sources-dependent variable subsection) on how clients for exit interviews were randomly selected in the current version. 

“If there was more than one client exiting the facility, the first client to be interviewed was randomly selected by using a lottery system. In this system, each client was assigned a number and thereafter each number was written on each piece of paper. All papers were shuffled well and one paper was picked randomly to obtain the first interviewee. After returning from an interview, the data collector selected the next patient entering the consultation room until the targeted sample size was achieved.” The reference for the methodology is provided as well.

List of references could be shortened.

We have refined some references

Reviewer #2: DEAR Author,

The study’s topic, “Clients satisfaction at primary healthcare facilities and its association to the status of client service charter implementation in Tanzania” it’s interesting, but some sections should be clarified.

-The methodology is unclear, such measure tool or scare lack references support, and lack reliability and validity, and “Explain the purpose of this 5-minute interview to the client…” how to mention the Questionnaire could be measured.

We have added some references to support our methodologies. Details on how the exit interview was conducted have been added as well. 

-In addition, the study’s satisfaction score points present dichotomy, only yes or no, why result could be presented percentage.

“Yes” and “No” were the possible values for each question. To obtain the proportion (in percentage) of the facilities scoring yes for each question; the number of facilities scoring “yes” was the numerator and the total number of facilities included in the analysis was the denominator. 

-Result section, Figure 2: the satisfaction with services data why lower more than 80%, suggest author address more.

Yes, 72.2% of PHC facilities had their clients satisfied with services provided on a day of assessment. This finding is below the target value of 80%.

-“Table 4. Predictors of clients’ satisfaction at healthcare facilities during SRA of 2017/18.” incorrectly presented, samples could not match the study’s text, please check.

Some corrections have been made. 

-For more contributions, suggest authors should clarify each clients satisfaction item and analyze the samples which clients satisfaction item is more meaningful to be improved. Such as patient characteristics or facilities health workers’ quality care.

-The discussion section should be reworded based on the incorrect analysis result.

Thank for the comment. We have improved some of the paragraphs in this section.

-Page 18, “Second, the coronavirus disease of 2019 (COVID-19) has exacerbated the already weak health systems in low- and middle- income countries threatening to disrupt services…” it does not include the study range, so I am confused because page 11 authors presented “….By 2017 Tanzania had 7,289 PHC facilities that were all involved in SRA assessment of the year 2017/2018.”

It is true the results of this study were obtained before the onset of the covid-19 pandemic. However, given the current situation in Tanzanian facilities and what we have learned after the covid-19 outbreak, we thought it was good to advise the government and stakeholders what could be the implication of the past findings in the current situation. And therefore they need to put in place robust systems to protect health care consumers from merging and re-emerging outbreaks/harm in the future.

Thank you for your effort.

Reviewer #3: Introduction

Those who are not familiar with primary healthcare do not really know what the Astana Declaration is. Please elaborate a bit more in detail.

Details on the declaration has been added. 

Patient satisfaction in primary care is a controversial outcome as international studies have shown ceiling effects. Please discuss.

A ceiling effect is said to occur when a high proportion of subjects in a study have maximum scores on the observed variable. We have discussed this as one of the limitations to our findings in manuscript text. The reference has been added as well.

On page 4 it is stated „This paper discusses service area number seven (7) of the star rating assessment (SRA) tools, namely client focus.“ without explaining the SRA tools. Therefore, the reader is not able to follow the context.

We have added a description on the SRA tool to help readers get a link between SRA tool, a star award and area 7 which is client focus. 

It is not clear why the SRA dataset of 2017/18 was used. Why was there no newer data?

SRA dataset of 2017/18 is the newest nation-wide dataset so far. We have just started the third phase of SRA this year (2021/2022) by reaching out 10 among 26 regions available in the country. This is very expensive task and we are not sure yet if we will complete the task to all regions. The first SRA was conducted in 2015/16, however, it was not possible to maintain a quality database by that time. 

Methods

In the Conceptual Framework the details of the content and implementation of the client service charter should be described. Is there a SOP which describes how to implement the CSC? How is it measured if a CSC is implemented or not?

As a country we have a guideline titled “National Client’s Service Charter for Health Facilities” in which chapter 4 describes on how to monitor and evaluate how facilities perform on implementing CSC.

Monitoring is conducted on quarterly basis through customer satisfaction surveys. The same tool that was used to collect data for this study is used for the surveys. 

„due to one reason or another, facilities whose results were not found in the database were excluded from the analysis.“ => These reasons have to be explained in detail. Otherwise you have a sample with unknown selection criteria which influence your results.

The statement has been modified to make it clearer, thank you so much for the observation. The previous statement was not presenting the actual thing we did. We targeted all facilities that were assessed through Star Rating Assessment (SRA). However, some had missing data during data extraction and were excluded from the analysis.

Table 3: „Client service charter was implemented if all three indicators were met.“ Does this mean that all the displayed questions in Table 3 had to be answered with „Yes“ ? Please explain.

Thank you for the observation. To be more precise, the statement has been modified to “The indicator for the client service charter was regarded implemented if all verification questions scored “Yes” for that particular indicator”

The Analysis section has to be much more detailed. Statistical measures used have to be justified and described. The binary variable regarding patient satisfaction should be explained more clearly. Those who scored 8/10 were considered to be satisfied with the service?

More details have been added in different sub-sections of the Methodology section. Specifically, clarification on a binary variable has been made. 

Results

There are 2766 (38%) PHC facilities not included in the analyses. A missing analysis regarding their characteristics has to be performed in order to demonstrate if you have a selective sample or not.

A sensitivity analysis was done and realized that excluded data had similar characteristics to those that were included. To clarify more, we have included a table numbered 4 that describe the characteristics of missing facilities. 

Table 4: (currently Table 5)

As far as I have understood the authors have performed single-variable regressions first with each of the listed predictors and then included them all in one multivariate analyses. Please explain more precisely. Please give an indicator of model performance, e.g. McFadden´s R-square.

Thank you for the nice comment. However, there is a mixing understanding of the rationale for reporting indicators of model performance. For example, Giselmar et al (https://doi.org/10.1177/0049124116638107 ) in their study explain the following “the McFadden values are not appropriate in cases with large sample sizes (n > 200) and/or strongly asymmetric distribution of observations to categories (ν > 1.6)”

Therefore, we opted to include not the values to avoid potential confusion. However, we are ready to comply with the reviewer’s comment if he/she insists.

Discussion

Important points were raised for the future of the PHC system in Tanzania. Please explain further the discrepancy between the relatively low implementation of client service charter and the relatively high satisfaction of patients.

Thank you for the comment. We have added explaination in discussion and third paragraph of conclusion section to explain further the discrepancy between the relatively low implementation of client service charter and the relatively high satisfaction of patients.

Reviewer #4: The logical comments are as below.

1. The authors defined "Three clients were randomly selected at each dispensary and five at health centers and hospital level 1" as the facilities clients was satisfied with, and potential factors were examined for the associations with the client-satisfying facility. Three or five clients are not enough to evaluate the facilities in terms of client satisfaction. The authors should analyze the associations between client satisfaction and factors, not between facilities evaluated by a limited number of clients and factors.

We agree with the reviewer; three to five clients are few to evaluate client satisfaction at the facility. However, However, the experience during data collection showed the number of clients attending the majority of facilities i.e. dispensaries was very low to the extent it was difficult in some instances to achieve the required number of interviewees. The findings of this study are presented as “satisfaction on a day of the visit” and should be translated with such a precaution. We have included this as one of the limitations of the study in the manuscript text.

2. The authors did not describe the method of random selection of clients. The readers cannot judge whether the clients were sampled randomly or not.

We have added a description (in manuscript text) on how clients for exit interviews were randomly selected in the current version. 

“If there was more than one client exiting the facility, the first client to be interviewed was randomly selected by using a lottery system. In this system, each client was assigned a number and thereafter each number was written on each piece of paper. All papers were shuffled well and one paper was picked randomly to obtain the first interviewee. After returning from an interview, the data collector selected the next patient entering the consultation room until the targeted sample size was achieved.”

3. The characteristics of clients were not described. The readers cannot understand whose evaluation was the base of satisfaction.

This is one of the limitations of the study, we have it in the manuscript text. The database we used does not contain the characteristics of the clients interviewed.

4. The characteristics and qualification of interviewers were not described.

We have added a special sub-section named “data sources” under “Study variables and collection of data”. In the second paragraph (with the reference), we have described the characteristics and quality of data collectors.

5. Figure 2 is hard to be interpreted. The title says "Proportion of primary health facilities whose clients are satisfied". For example, what did the author mean for the proportion of "Charter displayed?"?

The figure has been revised both structurally and descriptive to make it clearer. Thank you for the comment.

The technical comments are as follows.

1. The abbreviation must be defined at the first appearance with the full spelling, and the abbreviation should be used thereafter. For example, "CSC".

The document has been reviewed and corrections made as per reviewer’s comment.

2. In Table 2, delete "=" after "Wait less than 60 minutes".

The symbol has been omitted, thank you for the observation.

3. In Table 4, COR of 1.125 should be replaced with "1.13". The term "Ref" is dropped at four parts.

Thank you for the comment, changes have been made.

Reviewer #5: Dear authors

I would like to thank you for giving me the opportunity to review this well-written manuscript entitled “Clients satisfaction at primary health-care facilities and its association to the status of client service charter implementation in Tanzania”. The main objective of the study was to determine clients’ satisfaction at PHC facilities and its association with the status of client service charter implementation in Tanzania. This work is well designed and well well-written script that can improve the delivering PHC services. Therefore, I think the manuscript can be considered seriously for publishing. I have just one comment:

1. I would like to see some information regarding Tanzania such as population, socio-economic status, and PHC in the introduction.

Best wishes

I thank the reviewer for the positive comments. The recommendations have been incorporated in the introduction section.

---

## [Decision Letter · Decision Letter 1]

18 Jul 2022

Clients satisfaction at primary healthcare facilities and its association with implementation of client service charter in Tanzania

PONE-D-21-39135R1

Dear Dr. Erick S. Kinyenje,

We’re pleased to inform you that your manuscript has been judged scientifically suitable for publication and will be formally accepted for publication once it meets all outstanding technical requirements.

Kind regards,

Wen-Wei Sung, M.D., Ph.D.

Academic Editor

PLOS ONE

Reviewers' comments:

Reviewer's Responses to Questions

**Comments to the Author**

1. If the authors have adequately addressed your comments raised in a previous round of review and you feel that this manuscript is now acceptable for publication, you may indicate that here to bypass the “Comments to the Author” section, enter your conflict of interest statement in the “Confidential to Editor” section, and submit your "Accept" recommendation.

Reviewer #2: All comments have been addressed

Reviewer #5: All comments have been addressed

2. Is the manuscript technically sound, and do the data support the conclusions?

Reviewer #2: Partly

Reviewer #5: Yes

3. Has the statistical analysis been performed appropriately and rigorously? 

Reviewer #2: Yes

Reviewer #5: Yes

4. Have the authors made all data underlying the findings in their manuscript fully available?

Reviewer #2: Yes

Reviewer #5: Yes

5. Is the manuscript presented in an intelligible fashion and written in standard English?

Reviewer #2: Yes

Reviewer #5: Yes

6. Review Comments to the Author

Reviewer #2: Dear Author,

Thanks for your work. My part that you had response my concerns, I have no other suggestion.

Reviewer #5: I want to thank you for giving me the opportunity to review the revised version of this manuscript. The authors addressed my comment sufficiently.

7. PLOS authors have the option to publish the peer review history of their article (what does this mean?). If published, this will include your full peer review and any attached files.

Reviewer #2: No

Reviewer #5: **Yes: **Abbas Mardani

---

## [Editor Report · Acceptance letter]

5 Aug 2022

PONE-D-21-39135R1 

Clients satisfaction at primary healthcare facilities and its association with implementation of client service charter in Tanzania 

Dear Dr. Kinyenje:

I'm pleased to inform you that your manuscript has been deemed suitable for publication in PLOS ONE. Congratulations! Your manuscript is now with our production department. 

Kind regards, 

on behalf of

Dr. Wen-Wei Sung 

Academic Editor

PLOS ONE